# Crop, Host, and Gut Microbiome Variation Influence Precision Nutrition: An Example of Blueberries

**DOI:** 10.3390/antiox12051136

**Published:** 2023-05-22

**Authors:** Connie M. Weaver, Mario G. Ferruzzi, Maria Maiz, Dennis P. Cladis, Cindy H. Nakatsu, George P. McCabe, Mary Ann Lila

**Affiliations:** 1School of Exercise and Nutritional Science, San Diego State University, San Diego, CA 92182, USA; 2Arkansas Children’s Nutrition Center, Department of Pediatrics, University of Arkansas for Medical Sciences, Little Rock, AR 72202, USA; mferruzzi@uams.edu; 3Department of Nutrition Science, Purdue University, West Lafayette, IN 47907, USA; m.maizr@gmail.com; 4Department of Food Science and Technology, Virginia Polytechnic Institute and State University, Blacksburg, VA 24061, USA; dcladis@vt.edu; 5Department of Agronomy, Purdue University, West Lafayette, IN 47907, USA; cnakatsu@purdue.edu; 6Department of Statistics, Purdue University, West Lafayette, IN 47907, USA; mccabe@purdue.edu; 7Plants for Human Health Institute, Department of Food Bioprocessing and Nutrition Sciences, North Carolina State University, Kannapolis, NC 28081, USA

**Keywords:** blueberries, polyphenols, anthocyanins, precision nutrition

## Abstract

Epidemiological studies have shown associations between polyphenol-rich fruit intake and bone health, and preclinical studies have shown that blueberries improve bone health. To determine the genotype and dose of blueberries that are effective in ameliorating age-related bone loss, a multi-institutional team of investigators performed in vitro, preclinical, and clinical studies on blueberry varieties that differed in flavonoid profiles. Principal component analysis was used to select blueberry genotypes that varied in anthocyanin profiles. Total phenolic content did not predict the bioavailability of polyphenolic compounds in rats. A range in bioavailability was observed in individual polyphenolic compounds across genotypes. Both alpha and beta diversity analyses indicated that gut microbiome profiles varied with blueberry dose in rats. Additionally, the identification of specific taxa, such as Prevotellaceae_UCG-001 and Coriobacteriales, increasing after blueberry consumption adds to the mounting evidence of their role in polyphenol metabolism. All of the sources of variation can inform blueberry breeding practices to influence precision nutrition.

## 1. Introduction

Precision nutrition is of great current interest and is a growing focus of nutrition research funding. The concept is aimed at predicting an individual’s response to a food or diet to make personalized dietary recommendations for optimal health and disease prevention for that individual. This contrasts with traditional dietary guidelines for public health recommendations that are framed for generally healthy people. However, evaluating individual diversity in response to diet is just one part of the complexity that is involved. Diversity occurs along the whole spectrum, from the food, beverage, or supplement being consumed to the physiology of the individual consuming the food and the spectrum of metabolites generated by that individual’s gut microbiome and host metabolizing systems.

Fruit and vegetable consumption is associated with a decreased risk of developing many chronic diseases, including obesity, diabetes, cardiovascular disease (CVD), and neurocognitive diseases [1,2,3,4]. These benefits are commonly attributed to bioactive phytochemicals produced by plants as non-nutritive, secondary metabolites that possess biological activities in humans, including antioxidant, anti-inflammatory, anti-cancer, antimicrobial, and neuroprotective effects [5,6,7]. Of the thousands of phytochemicals that have been identified, some of the most promising health benefits have been linked to (poly)phenolic (hereafter referred to as polyphenolic) compounds associated with tea, cocoa, and especially berry fruits [8,9]. These include phenolic acids, stilbenoids, and flavonoids (including anthocyanins).

Blueberries (*Vaccinium* spp.) are a well-recognized source of dietary polyphenols. Anthocyanins account for nearly half of all polyphenols present in blueberries and give them their distinctive blue-purple color [10]. Evidence of health benefits from epidemiological studies, coupled with the abundance of polyphenols, has catalyzed further preclinical and clinical research to elucidate the mechanisms of action and substantiate knowledge on the health benefits attributed to blueberries and blueberry-rich foods.

Intervention trials support the ability of blueberries to improve markers of cardiovascular function (e.g., blood pressure, arterial blood flow, and cholesterol levels) while decreasing the risk of developing CVD [11,12]. Additionally, studies regarding neurocognition, memory, and slowing age-related cognitive decline indicate that blueberries positively influence these endpoints [13,14]. Further studies with anthocyanins have shown similar results and worked through the same mechanisms, suggesting that anthocyanins are likely one of the bioactive agents responsible for many of the health benefits observed with blueberries [15,16]. Although all blueberries are a rich source of polyphenols, there are hundreds of genotypes available to farmers and consumers, all with varying polyphenolic profiles, ranging in concentration of total polyphenolic (TP) content from 1.63 to 2.87 mg gallic acid equivalents (GAE)/g of fresh weight (FW) [17], demonstrating potential for differences in polyphenol absorption, biotransformation, and utilization. The differences observed in polyphenolic content are due to plant genetics, environmental factors, and agronomic conditions, as well as the type and extent of post-harvest storage and processing [18,19,20].

A multi-institutional team of investigators funded by the National Institutes of Health performed in vitro, preclinical, and clinical studies on blueberries that differed in polyphenolic profiles for bone health outcomes. This paper reports the results from this coalition, demonstrating the diversity of polyphenol profiles in blueberries, the bioavailability of anthocyanins from blueberries selected for their diversity in blueberry profiles, and the influence of blueberries on the gut microbiome. This work provides context for understanding the complexity involved at all stages, from plant genetics to host to the gut microbiome pathway that impacts health.

## 2. Materials and Methods

### 2.1. Diversity of Anthocyanin Profiles across Blueberry Genotypes

Anthocyanin profiles were available from 267 genotypes of blueberries from commercial genotypes and breeding selections at the North Carolina State University Piedmont Research Station (NCSU-PRS), Salisbury, North Carolina, USA [21]. Principal Component Analysis (PCA), a multivariate statistical technique [22], was performed on 17 anthocyanins profiled in each genotype.

### 2.2. Diversity of Blueberry Polyphenolic Profiles

Genotypes with sufficient diversity in their anthocyanin—and, by extension, their polyphenolic—profiles, based on selecting divergent profiles from the PCA, were selected for characterizing their polyphenolic profiles before determining the bioavailability of flavonoids among blueberry genotypes in ovariectomized rats as a model for bone loss in postmenopausal women.

#### 2.2.1. Blueberry Material

Six genotypes from the NCSU-PRS collection: three rabbiteye genotypes (*Vaccinium virgatum*), including Ira, Montgomery, and Onslow, and three southern highbush genotypes (*V. corybosum*), including Legacy, Sampson, and SHF2B1-21:3, a clone developed by the NCSU breeding program [17], were selected. To maximize the diversity of polyphenolic profiles, four additional members of the *Vaccinium* genus, including bilberries (*V. myrtillus*, Oregon Wild Harvest, Sandy, OR, USA), cranberries (*V. macrocarpon*, Cranberry Network, Wisconsin Rapids, WI, USA), and lowbush blueberries (*V. angustifolium*) from both a single field collection of wild blueberries in Ellsworth Maine (designated as Wild Blueberries) and a commercially prepared wild blueberry composite from all major growing regions in Maine and maritime Canada designated as a Lowbush Blueberry (LB) composite (Wild Blueberry Association of North America, Old Town, ME, USA), were also analyzed. All animal studies used LB composite in the treatment arm.

#### 2.2.2. Quantification of Individual Blueberry Phenolics

Lyophilized whole blueberries were ground into a fine powder using a spice grinder. Phenolics were then extracted, purified, and analyzed in triplicate for total phenolics via the Folin–Ciocalteu method, as described elsewhere [23,24]. Individual phenolics in purified blueberry extracts were filtered with 0.45 µm PTFE filters and 10 µL was injected into a Waters 2695 LC-MS (Milford, MA, USA) with a Waters X-Bridge BEH Shield RP18 column (2.5 µm, 2.1 × 100 mm) heated to 40 °C as previously described [23,25]. Anthocyanins were analyzed using a linear gradient at a flow rate of 0.25 mL/min, with mobile phases A (2% formic acid in water) and B (0.1% formic acid in acetonitrile) in the following ratios at 0, 15, 18, and 19 min, respectively: 90:10, 75:25, 70:30, and 90:10. Phenolic acids, stilbenoids, flavan-3-ols, and flavonols were analyzed using a linear gradient at a flow rate of 0.25 mL/min, with mobile phases A (0.4% formic acid in water) and B (0.1% formic acid in acetonitrile) in the following ratios at 0, 15, 17, and 18 min, respectively: 95:5, 65:35, 50:50, and 95:5.

Phenolics were detected and quantified using a Waters Micromass ZQ 2000 mass spectrometer, as shown in Table 1. Positive mode electrospray ionization (ESI+) was used for detection and quantitation of anthocyanins, with the following settings: source temperature 150 °C, desolvation temperature 250 °C, nitrogen desolvation gas flow rate 250 L/h, nitrogen cone gas flow rate 25 L/h, capillary voltage 2500 V, cone voltage 50 V, and extractor voltage 3 V. Negative mode ESI (ESI-) was used to detect and quantify phenolic acids, stilbenoids, flavan-3-ols, and flavonols, with the following settings: source temperature 150 °C, desolvation temperature 250 °C, nitrogen desolvation gas flow rate 400 L/h, nitrogen cone gas flow rate 60 L/h, capillary voltage 3000 V, cone voltage 40 V, and extractor voltage 5 V. Total monomeric anthocyanins were quantified using the pH differential method [26] and expressed in cyanidin-3-glucoside equivalents.

#### 2.2.3. Statistical Analysis

Polyphenolic characterization from blueberry raw material data are presented as mg polyphenol/100 g of berries (dry weight).

### 2.3. Diversity of Bioavailability of Blueberry Polyphenols—Acute PK Study

All animal studies were conducted under the guidance of, and with protocols approved by, the Purdue University Animal Care and Use Committee (Protocol 1405001076). Twenty 3-month-old rats were ovariectomized (as a model for postmenopausal women) and, upon arrival, individually housed in stainless steel wire-bottom cages with a 12 h on:off light cycle and fed a chow diet and deionized water ad libitum. Rats were acclimated to their new housing and stabilized after their surgery for two months. Nine days prior to the pharmacokinetic study, rats were put on a polyphenol-free diet (modified AIN93M diet where soybean oil was replaced with corn oil) and randomized to one of the five blueberry genotype treatment groups: Montgomery, Ira, Onslow, SHF2B1-21:3, and LB composite. The study design is shown in Figure 1.

Whole, freeze-dried blueberries were milled to a ~0.5 mm particle size. TP content was used to determine the individual dose of blueberry for each rat. Each dose was normalized to provide 25 mg TP/kg of body weight (BW). Individual doses were suspended in 2 mL of ultrapure water, vortexed, and administered to the rats on the day of the pharmacokinetic study via oral gavage.

Two days prior to the study, rats were anesthetized with isofluorane (3–5%) in an anesthesia chamber and maintained under anesthesia with a mask (1.5–3% isofluorane). A silastic catheter was implanted into the right jugular vein and they were given buprenorphine (0.1 mg/kg BW) to alleviate pain prior to regaining consciousness. Rats were allowed to rest for 48 h after surgery and catheters were kept patent by flushing with heparinized saline (20 U/mL) every 12 h. Rats were fasted 8 h prior to the study and ~400 µL of whole blood was collected at baseline (prior to gavage with their respective blueberry dose) and 0.25, 0.5, 1, 2, 4, and 8 h post-gavage from the jugular catheter into lithium-heparinized tubes. Whole blood was processed to plasma by centrifugation at 4 °C and 6000 rpm for 10 min. Two aliquots of 80 µL of plasma each were combined with 20 µL of acidified saline (1% ascorbic acid (AA), *w*/*v* %), flushed with N2 gas, and stored at −80 °C until analysis. Rats were euthanized at the end of the 8 h pharmacokinetic study by carbon dioxide asphyxiation. Uterine horns were excised and weighed to confirm ovariectomy.

#### 2.3.1. Analysis of Phenolic Metabolites from Plasma

Anthocyanin, flavonols, and flavan-3-ol metabolites were extracted from plasma via solid-phase extraction (SPE) as described by Chen et al. [27]. Briefly, samples were loaded into preconditioned SPE cartridges. The cartridges were then washed with 2 mL of 2% formic acid (FA) (aq.) for anthocyanins and 1.5 M FA (aq.) for flavonols/flavan-3-ols, followed by 1 mL 5% MeOH (% *v*/*v*). Anthocyanins were eluted with 2 mL of 2% FA MeOH (% *v*/*v*) and flavonols/flavan-3-ols were eluted with 2 mL of 0.1% FA MeOH (% *v*/*v*). Samples were dried down with N2 gas. Dried extracts were resuspended in LC mobile phases and injected into the LC-MS/MS.

Analysis of plasma anthocyanin, flavanols, and flavan-3-ol metabolites was performed as described by Chen et al. [27] using an Agilent 6400 triple quadrupole (QQQ) mass spectrometer equipped with an electrospray ionization (ESI) source under multiple reaction monitoring modes (MRM, Table 1). Briefly, separation and characterization of metabolites was achieved using an XBridge BEH Shield RP-C18 column (2.5 µm, 2.1 × 100 mm). For (+)catechin/epicatechin, quercetin, and myricetin metabolites, binary mobile phases were A: 0.1% FA (aq.) (% *v*/*v*) and B: 0.1% FA in acetonitrile (% *v*/*v*). The column was heated to 30 °C and the system flow rate was 0.3 mL/min. The binary solvent gradient was: 10% B at 0 min, 40% B at 10 min, 95% B at 11 min, and back to 10% B at 12–18 min. Fragmentor voltage was set at 135 V and collision energy was 17 eV in negative ESI for all compounds.

For anthocyanin metabolites, binary mobile phases were A: 2% FA (aq.) (% *v*/*v*) and B: 0.1% FA in acetonitrile (% *v*/*v*). The column was heated to 30 °C and the system flow rate was 0.3 mL/min. The binary solvent gradient was: 5% B at 0 min, 25% B at 15 min, 30% B at 19 min, and back to 5% B at 19–22.5 min. Fragmentor voltage was set at 135 V and collision energy was 17 eV in positive ESI for all compounds.

#### 2.3.2. Statistical Analysis

Data from the bioavailability study were expressed as mean ± standard error of the mean (SEM). Pharmacokinetic parameter area under the curve from 0 to 8 h (AUC_0–8 h_) was calculated via the linear trapezoidal method using Microsoft Excel 2013 v.15. Maximum plasma concentration (C_max_) and the time of the maximum plasma concentration (T_max_) were obtained from the pharmacokinetic curve. Both AUC_0–8 h_ and C_max_ were normalized by dose of parent anthocyanin, flavonol, and flavan-3-ol present within each individual dose of blueberry treatment. Statistical analyses were performed using JMP 17.0 and SAS 9.3 (SAS Institute, Cary, NC, USA) for bioavailability and phenolic characterization, respectively. Data were analyzed via two-way ANOVA and one-way ANOVA for phenolic characterization and bioavailability, respectively. Significant differences were evaluated via the Tukey–Kramer HSD post hoc pairwise comparison test. Significance was accepted at α < 0.05.

### 2.4. Dietary Blueberry Effects on the Gut Microbiome—Repeat Dose, Crossover

#### 2.4.1. Study Design

Microbiome analysis was performed on 4-month-old female ovariectomized Sprague Dawley rats (Harlan Laboratories) designed to determine the effect of blueberries on bone calcium retention as previously described [28]. Briefly, twenty 5-month-old ovariectomized Sprague Dawley rats were put on a polyphenol-free diet to determine baseline to determine baseline measures then randomized to four 10-day treatment periods on a LB composite-enriched diet (2.5%, 5%, 10% or 15% LB composite diet). Each treatment was followed by a 10-day washout period on a polyphenol-free diet. Fecal samples for microbiome analysis were collected on day 10 of each of the treatments and washout periods and stored at −80 °C until analysis.

#### 2.4.2. Microbiota Extraction and Analysis

Fecal DNA extraction and amplicon sequencing. DNA for microbiota analysis was extracted from 160 fecal samples collected from 20 rats at the end of baseline, ethe nd of each of the 10-day blueberry treatments, and the end of each washout phase. DNA was extracted using the FastDNA SPIN Kit for Soil (MP Biomedicals) following the manufacturer’s instructions, and DNA quality and quantity were determined as previously described [29]. PCR amplification of the V3-V4 (343F: TACGGRAGGCAGCAG to 804R: CTACCRGGGTATCTAATCC) region of the 16S rRNA gene was performed as previously described [29] and sequenced using MiSeq Illumina 2 × 250 paired-end sequencing.

Sequence analysis. Sequences were analyzed using the QIIME2 pipeline [30] (version 2-2022.11). Sequences were paired, trimmed for quality, and sorted into representative sequences (amplified sequence variants, ASV) using DADA2 [31]. Representative sequences were assigned taxonomies using the q2-feature-classifier [32] trained using the SILVA dataset [33] (version 138_99). ASVs were filtered from the dataset if they represented less than 0.1% frequency of the mean sample depth, were found in only two samples, matched chloroplast or mitochondria sequences, or had non-assigned taxonomies. For an equal number of reads for alpha and beta diversity analyses, sample sets were rarified to 4300 reads. Alpha diversity was analyzed using Faith’s phylogenetic diversity (PD), Shannon, observed features, and Pielou’s evenness. Beta diversity was analyzed using both phylogenetic Unifrac distances [34] (weighted and un-weighted) and non-phylogenetic distances (Jaccard and Bray–Curtis).

#### 2.4.3. Microbiota Statistics

For all analyses, microbiome data from baseline and washout phases were combined to represent the 0% blueberry treatment. Data were first tested for normal distribution using the Shapiro–Wilk W test. Statistical differences of alpha-diversity measurements among all groups were determined using the Kruskal–Wallis test, as well as pairwise comparisons followed by Benjamini–Hochberg FDR correction. Statistical differences among beta-diversity measures among communities were determined using 999 permutations of PERMANOVA [35] and pairwise comparisons with Benjamini–Hochberg FDR correction. PERMDISP (permutational analysis of multivariate dispersions) was performed to ensure significance was not due to dispersion differences among samples [36]. Differences were considered significant if *p* < 0.05. Differential abundances in taxonomy across all samples were determined using Analysis of Composition of Microbiomes (ANCOM) [37].

## 3. Results

### 3.1. Diversity of Anthocyanin Profiles across Blueberry Genotypes

Raw data available for 17 anthocyanins, measured in mg/100 g wet weight (frozen fruit) for 1086 individual plants from 267 genotypes of blueberries [17], were used for analysis to identify priority genotypes for study that represent the broader diversity of the blueberry germplasm. In this instance, measures of plants from the same genotype were averaged. The anthocyanins were cyanidin 3-*O*-arabiniside, cyanidin 3-*O*-galactoside, cyanidin 3-*O*-glucoside, cyanidin 6-*O*-glucoside, delphinidin 3-O-arabiniside, delphinidin 3-*O*-galactoside, delphinidin 3-*O*-glucoside, delphinidin 6-*O*-glucoside, malvidin 3-*O*-arabiniside, malvidin 3-*O*-galactoside, malvidin 6-*O*-galactoside, malvidin 3-*O*-glucoside, malvidin 6-*O*-glucoside, peonidin 3-*O*-galactoside, petunidin 3-*O*-arabiniside, petunidin 3-*O*-glucoside, and petunidin 6-*O*-glucoside. Anthocyanins that were not normally distributed were log-transformed.

Figure 2 is a plot of the values of the second standardized principal component versus the first standardized principal component for our 267 genotypes of blueberries. The mean and standard deviation of the total concentration of the 17 anthocyanins were 17.30 and 6.03, respectively. The correlation of the total with the first principal component was 0.93, indicating this component essentially represents the total. The five principal points are marked with a P and genotypes selected for further study were identified within a 0.2 region of each of these points. Final genotype selection also depended upon available supply to meet the design for animal and human studies.

### 3.2. Diversity of Blueberry Polyphenolic Profiles

#### Polyphenol Content of Blueberries Used in Bioavailability Studies

Using the 6 blueberry genotypes with divergent anthocyanin profiles from the PCA analysis (Figure 2) and the 4 additional members of the Vaccinium genus, total phenolic content (TP) ranged from 1951–4627 mg/100 g berry (DW) and total monomeric anthocyanins ranged from 369–1722 mg/100 g berry (DW), with bilberry and the LB composite having higher levels of phenolics and total monomeric anthocyanins than their highbush counterparts (Figure 3).

Anthocyanins were the most abundant phenolic class present in *Vaccinium* species, accounting for approximately half of all phenolics in blueberries (Table 2, Figure 4). Considerable variation in the amounts and proportions of different anthocyanins was present across the 10 genotypes we examined. Consistent with the higher levels of total phenolics and total monomeric anthocyanins observed in bilberry and LB composite, these same genotypes had the highest levels of anthocyanins. Bilberry had the highest levels of delphinidin and cyanidin species, and the LB composite had the highest levels of malvidin and acylated anthocyanins. Cranberry had a significantly different anthocyanin profile compared with other berries, with the highest levels of peonidin but the lowest levels of delphinidin, malvidin, and petunidin species. Glycosylation also varied among the genotypes tested, with most exhibiting substantial amounts of arabinoside and galactoside derivates. Some genotypes (e.g., Ira, Legacy, and Sampson) had very low levels of glucosidic derivatives, while other genotypes (e.g., Onslow, SHF2B1-21:3, WBB, bilberry, and LB composite) had levels of glucosidic derivatives equal to or greater than the arabinosidic and galactosidic derivatives.

The most prominent phenolics are presented in Table 3 and Table 4, with chlorogenic acid (20–755 mg/100 g berry (DW)) and quercetin (93–418 mg/100 g berry (DW)) exhibiting the highest concentrations in berries. Flavan-3-ols were present in comparatively low amounts for all genotypes. As with anthocyanins, cranberry had a very different phenolic profile than the other *Vaccinium* species evaluated here, with the lowest level of chlorogenic acid and the highest levels of epicatechin, quercetin-3-*O*-arabinoside, and myricetin-3-*O*-glycosides.

### 3.3. Diversity of Bioavailability of Blueberry Polyphenols

#### Bioavailability of Blueberry Metabolites

Anthocyanin metabolites detected circulating in plasma after an acute dose to OVX rats included cyanidin-3-*O*-glycosides (Cy-3-glcs), delphinidin-3-*O*-glycosides (Del-3-glcs), malvidin-3-*O*-glycosides (Mal-3-glcs), peonidin-3-*O*-glycosides (Peo-3-glcs), and petunidin-3-*O*-glycosides (Pet-3-glcs). The pharmacokinetic curves and pharmacokinetic parameters AUC_0–8 h_, C_max_, and T_max_ for all berry genotypes can be found in Appendix A. Of the anthocyanin metabolites, Cy-3-glcs and Mal-3-glcs from the Montgomery blueberry had significantly higher bioavailability in comparison to the other genotypes of berries (*p* < 0.01) (Figure 5A). For Del-3-glcs, Montgomery had significantly higher bioavailability than SHF2B1-21:3 and Onslow (*p* = 0.025), but was no different than Ira and LB composite. The C_max_ of anthocyanins did not differ among genotypes, but when these values were normalized by dose, C_max_ for Cy-3-glcs was significantly higher in Montgomery than in Ira, Onslow, and SHF2B1-21:3, reaching a maximum concentration in plasma of 24.96 ± 3.72 nM/mg of Cy-3-glcs dose (*p* < 0.001). Montgomery showed a higher C_max_ normalized by dose for Del-3-gls and Mal-3-glcs, with 14.65 ± 2.16 nM/mg Mal-3-glcs dosed and 14.84 ± 3.55 nM/mg Del-3-glcs, than the other genotypes (*p* < 0.01), respectively. All anthocyanins showed a similar T_max_ between 0.25–1 h, regardless of blueberry genotype (Appendix A).

Bioavailability and pharmacokinetic curves for metabolites of flavan-3-ols, (+)-catechin and (-)-epicatechin, i.e., epicatechin-5-O-glucuronide (EC-glcr), catechin-5-O-glucuronide (C-glcr), 3′-*O*-methylepicatechin-5-glcr (MeEC-glcr) and 3′-*O*-methylcatechin-5-glcr (MeC-glcr) are shown in Figure 5B and Appendix A, respectively. As shown in Appendix A, Ira reached a significantly higher plasma concentration of 0.040 ± 0.006 µM of C-glcr in comparison to LB composite, which only reached a C_max_ of 0.015 ± 0.005 µM (*p* < 0.05). Ira, Onslow, and Montgomery showed a higher AUC_0–8 h_ than SHF2B1-21:3 and LB composite, but when the AUC_0–8 h_ was normalized by dose to determine bioavailability, there were no significant differences between the AUC_0–8 h_ /dose of flavan-3-ols, regardless of blueberry genotype.

The concentration and pharmacokinetic curves for flavonols, i.e., quercetin-3-glu, are shown in Appendix A. Bioavailability for Q-glcr was significantly higher in Montgomery (1.191 ± 0.342 µM × h/mg Q) than SHF2B1-21:3 and LB composite (0.355 ± 0.05 and 0.424 ± 0.051 µM × h/mg Q, respectively) (*p* < 0.001), but was not different compared to Onslow and Ira (Figure 5C). Although the C_max_ for Q-glcr did not vary among blueberry genotypes, when normalized by dose, Montgomery reached a significantly greater C_max_ of 0.66 ± 0.16 µM/mg Q than Ira, SHF2B1-21:3, and LB composite, at 0.046 ± 0.018, 0.215 ± 0.026, and 0.14 ± 0.02 µM/mg Q, respectively (*p* < 0.05) (Appendix A). Montgomery also showed higher bioavailability of MeQ-glcr than Ira. T_max_ did not differ among blueberry genotypes for Q and MeQ, while AUC_0–8 h_, Cmax and Tmax for Myr-glcr did (Appendix A). Ira had a significantly higher bioavailability for myricetin (0.315 ± 0.038 µM × h/mg Myr) than for all other blueberry genotypes (*p* < 0.001) (Figure 5D). Montgomery and Onslow followed Ira with an AUC_0–8 h_/mg Myr of 0.168 ± 0.028 and 0.115 ± 0.005 µM/mg Myr, respectively. Both SHF2B1-21:3 and LB composite had a lower AUC_0–8 h_ of 0.001 ± 0.0003 and 0.014 ± 0.002 µM/mg Myr, respectively (*p* < 0.001). Regardless of the higher content of myricetin glycosides in SHF2B1-21:3 and LB composite, when normalized by dose, they exhibited a bioavailability that was significantly (*p* < 0.001) lower than the other blueberry genotypes. T_max_ was significantly higher for LB composite (*p* < 0.05).

### 3.4. Dietary Blueberry Effects on the Gut Microbiome

#### 3.4.1. Gut Microbiota

In total, 8,599,413 high-quality merged sequences were obtained after processing of MiSeq Illumina sequencing results by DADA2 and filtering out low-incident and non-target reads. Per sample, there was a median of 50,996.5 reads (range: 4321–134,668). To maintain all samples, 4300 reads were chosen by rarefaction for alpha and beta diversity analyses.

#### 3.4.2. Taxonomic Analysis

Throughout the study, composition of the microbiome was dominated by two phyla: the Firmicutes (mean 86.8 ± 6.0%) and Bacteroidota (mean 10.0 ± 4.8%). The overall Firmicutes-to-Bacteroidota ratio significantly decreased with increasing blueberry dosage (Kruskal–Wallis, *p* = 0.0001), as we saw in our previously reported 90-day toxicity study [28]. The highest ratios were in samples with no-blueberry diets (25.3 ± 111.1), followed by 2.5% blueberry (13.7 ± 17.2), 5% blueberry (11.8 ± 4.6), 10% blueberry (7.8 ± 4.2), and 15% blueberry (7.3 ± 2.8). Pairwise comparisons with Bonferroni correction indicated that ratios differed significantly between 0% blueberry and 10% (*p* = 0.015) and 15% (*p* = 0.008) blueberry, as well as between 5% blueberry and 10% (*p* = 0.006) and 15% (*p* = 0.008) blueberry.

#### 3.4.3. Alpha Diversity

Comparisons of the of gut microbiome diversity within each sample indicated a significantly higher diversity in samples with higher blueberry treatments using the Shannon diversity metric (measure of richness and evenness) (*p* = 0.001) (Figure 6) and Pielou’s evenness measure (*p* = 2.28 × 10^−10^). However, no significant differences in using the richness measures (i.e., number of taxa present) were observed when using Faith’s PD or observed features. This indicates that the change in diversity results from changes in the quantities of some taxa and not changes in specific taxa composing the communities.

#### 3.4.4. Beta Diversity

Comparison of microbial communities among the blueberry treatments also indicated significant differences using all four metrics tested (*p* = 0.001). The clearest distribution of samples was seen using a PCoA scatterplot of Bray–Curtis metrics (Figure 7), a measure comparing the presence, absence, and quantities of taxa present among the samples used. Samples separated along PCoA axis 1 accounted for 11.16% of variation.

Pairwise statistical comparisons indicated that 0% blueberry differed significantly from all the blueberry treatments (2.5, 5, 10, and 15%) (PERMANOVA, *p* = 0.001) and dispersion was not significantly different. As seen in the figure, pairwise comparisons of the 2.5 and 5% combined dosage communities and the 10 and 15% combined communities differed significantly from each other and 0% (q = 0.001), whereas in pairwise comparisons of each dose, the 2.5% blueberry treatment differed significantly from the 10 and 15% blueberry treatment, and the 5% treatment differed significantly from the 15% blueberry (figure not shown). Similar trends were found using the other beta diversity metrics tested, but the blueberry treatments separated along PCoA axis 2.

#### 3.4.5. Differentially Abundant Taxa

ANCOM identified specific taxa contributing to the differences among communities based on blueberry dosage treatment. ANCOM identified eight taxa that differed significantly, but on examination of the volcano plot of the data (Appendix A), only four of the taxa had high W (>100) and clr (>40) values. None of these taxa have been classified at the genus level. Two are from the phylum Actinobacteriota and listed as uncultured from the order Coriobacteriales and the other *incertae sedis*; one is from the phylum Bacteroidota family Prevotellaceae_UCG-001, and one from the Firmicutes family Anaerovoracaceae XIII_UCG-001. Boxplots of these taxa indicated that all their abundances were higher in samples after blueberry treatments (Figure 8).

## 4. Discussion

Blueberry was used as a model for studying the potential impact of crop genetic diversity on the bioavailability of polyphenolic constituents and variation in the gut microbiome, two factors associated with chronic disease outcomes.

### 4.1. Variation in Blueberry Anthocyanin Profiles

PCA, a statistical method that considers a large number of variables, such as anthocyanin profiles, for a large number of blueberry genotypes, allowed us to identify a small number of genotypes that effectively could serve as a way to capture the broad diversity present in the germplasm in further study. The ability to refine biomedical testing to a limited array of genotypes captures the broader diversity of the collection and allows for identification of bioactives unique to individual genotypes or common functions that can be generalized across the broader germplasm. Identifying functional traits with specific genotypes can inform breeding programs toward improving the nutritional quality of blueberries overall.

### 4.2. Variation in Bioavailability of Polyphenolics

Phenolic content of LB composite was twice that of highbush blueberries, consistent with previous studies [18]. This is likely explained both by genetics, as well as the response to the harsher environmental conditions where lowbush blueberries are grown. However, parent compounds are rarely observed circulating in plasma, suggesting that the bioactive compounds in blueberry are the multiple metabolites produced after first-pass metabolism that will eventually reach target tissues and exert their health benefits.

The anthocyanin metabolites for cyanidin and malvidin were significantly more bioavailable from the Montgomery genotype. Consistent with anthocyanin bioavailability studies in humans [38] and rats [39,40], all anthocyanins reached a maximal plasma concentration 0.25–1 h post-dose. A small part of anthocyanin absorption may begin early in the stomach, as evidenced in this study by the high concentration of circulating anthocyanin metabolites as early as the first blood draw at 15 min post gavage, but the greatest absorption occurred through the small intestine. The rate and extent at which they are absorbed may be affected by the aglycone, sugar moiety, or acylated groups [41]. The acylation of anthocyanins highly influences their bioavailability. Charron et al. showed that nonacylated anthocyanins were 4-fold and 4.7-fold more bioavailable than the acylated anthocyanins in red cabbage and purple carrot juice, respectively [42,43]. Only 5% of Montgomery’s total anthocyanin content were acylated compared to ~9–28% acylated anthocyanins for other genotypes.

Both flavan-3-ols and flavonols varied widely across genotypes. Once normalized for dose, there were few differences in flavan-3-ol pharmacokinetics. In contrast, the rabbiteyes, Montgomery, Ira, and Onslow had the lowest total quercetin content, but had higher bioavailability for quercetin glucuronide (*p* < 0.05) than the other berries. For its methylated conjugate, MeQ-glcr, Ira had the lowest bioavailability (*p* < 0.05) of all berries. This may be due to differences in the glycosylation of quercetin across the different blueberry genotypes. The sugar moiety highly influences quercetin bioavailability as it requires cleavage for full absorption by lactase phlorizin hydrolase (LPH) or cytosolic β-glucosidase (CBG) within the enterocyte. Once the aglycone is released, quercetin is then subject to host metabolism by phase II enzymatic conjugation systems including glucuronide/sulfates/methyl groups [44]. Except for glucose, all other glycosides are poor substrates for enzymatic cleavage, limiting the release of quercetin aglycone for absorption through the small intestine. Quercetin glycosides then travel to the lower gut, where they can be metabolized by lower gut bacteria [45]; this early and late absorption can be observed in Appendix A, where quercetin exhibits a pharmacokinetic curve with double peaks for its MeQ-gclr metabolite, which is not uncommon and is consistent with previous observations. Yang et al. reported a similar phenomenon with Q-glcr displaying peaks at 2 h and 8 h post dose, which, in their study, was possibly caused by enterohepatic recirculation [46]. In our study, the peaks for MeQ-glcr occurred at 0.5 h and 1–2 h post-dose, which could have been caused by an early and more rapid passive absorption of the quercetin aglycone in comparison to the slower active absorption of quercetin glucoside.

The rabbiteye genotypes Montgomery, Ira, and Onslow have a significantly lower concentration of myricetin in comparison to the highbush and lowbush genotypes, with a 23-fold difference in concentration between Ira and SHF2B1-21:3. Although Ira and Montgomery had significantly lower content of myricetin (*p* < 0.05), they reached significantly higher plasma concentrations of 7- and 11-fold higher than SHF2B1-21:3 (*p* < 0.05), respectively. The bioavailability from Ira was higher than all genotypes (*p* < 0.05), followed by Montgomery and Onslow, and a very low bioavailability was observed for SHF2B1-21:3 and LB composite. As with quercetin, myricetin absorption rate may be low because its absorption might have been influenced by the type of glycosidic linkages. Although the aglycone and glycosylated forms of myricetin were not differentiated in our LC method, it is possible that the differences in myricetin bioavailability could have been due to differences in content of myricetin aglycone and the glycosylated form.

### 4.3. Variation in Gut Microbiome in Response to Blueberry

Both alpha and beta diversity analyses indicated there were *dose-dependent* effects of blueberry on the gut microbiome in this rat model of postmenopausal estrogen deficiency. The increase in alpha diversity is suggested to be related to better microbiome gut health and to cognitive function [47]. Four taxa were found to be significantly higher in blueberry treatments compared to no blueberry controls, i.e., two Actinobacteriota and listed as uncultured from the order Coriobacteriales, one from the phylum Bacteroidota family Prevotellaceae_UCG-001, and one from the Firmicutes family Anaerovoracaceae XIII_UCG-001. The taxa listed as Prevotellaceae_UCG-001 and Coriobacteriales *incertae sedis* were found to be significantly associated with blueberry in the OVX mouse study described by Sato et al. [48]. The Coriobacteriaceae family are thought to be to polyphenol degraders [49].

The limitations of our studies varied. In the PCA, the sample size was large, but was limited in the bioavailability studies. Alternative approaches could lead to different conclusions. In the PCA analysis, clustering of genotypes did not consider categorical characteristics, such as type or region of origin. For studying bioavailability, we opted to give the same dose of TP (25 mg/kg BW), resulting in wide differences in the amount of freeze-dried blueberry powder dosed from 189 mg to 285 mg of blueberry powder. Genotype comparisons should be made for the gut microbiome, as was performed in our previous report of a mouse study [48].

## 5. Conclusions

Blueberries of different genetic backgrounds contain different phenolic profiles that vary in the extent of bioavailability and metabolism of their polyphenols. PCA of 17 anthocyanins from 267 genotypes allowed selection of a few genotypes with widely differing profiles for further characterization. The high bioavailability of anthocyanins and phenolics in certain genotypes is a unique contribution of this work. Further, evidence of a gut microbiome response to blueberry dose was observed. This rich variation encountered in each system from crop to consumer to gut microbiome can be used to improve crop selection, breeding practices, and identification of lead genotypes, which may be leveraged to understand functional responses for health and develop precision nutrition practices.

## Figures and Tables

**Figure 1 antioxidants-12-01136-f001:**
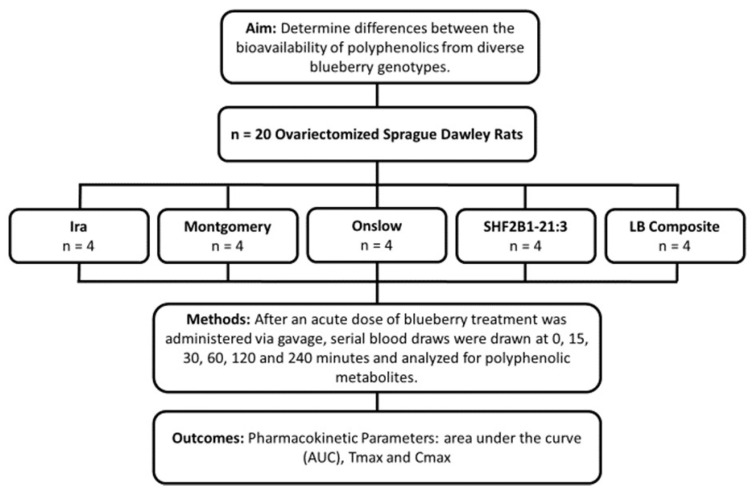
Study design for determining bioavailability of polyphenolics from blueberry genotypes in rats.

**Figure 2 antioxidants-12-01136-f002:**
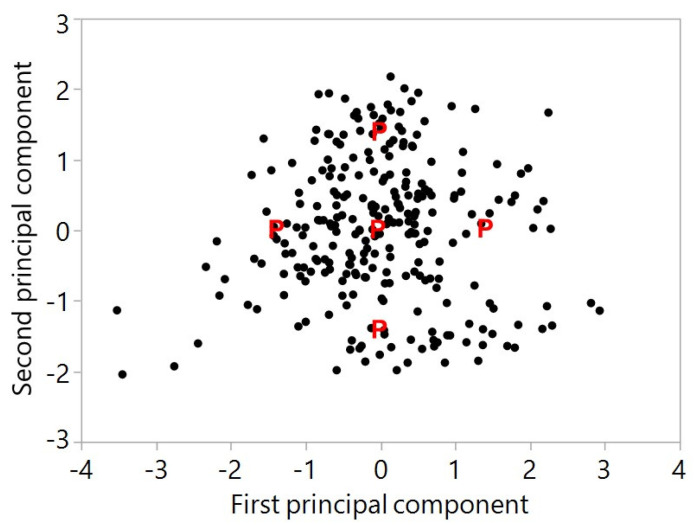
Principal component analysis displaying the distribution of 267 genotypes of blueberries and the principal points for selection of genotypes for further study on bioavailability.

**Figure 3 antioxidants-12-01136-f003:**
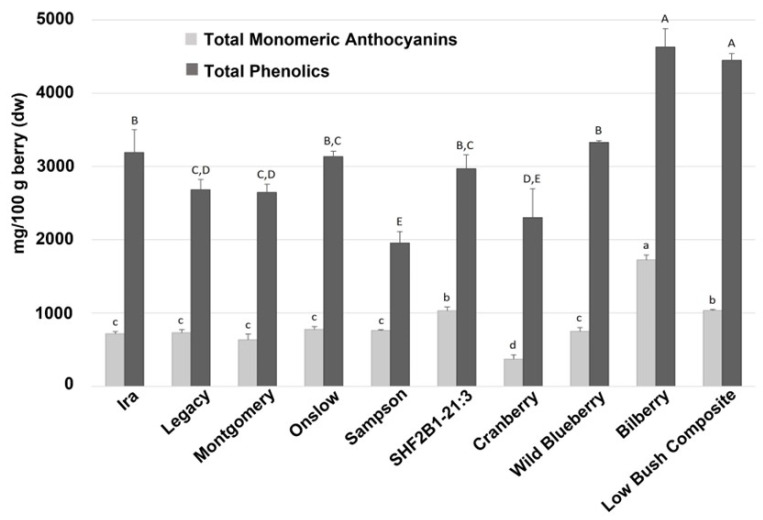
Total phenolics (in gallic acid equivalents) and total monomeric anthocyanins in 10 genotypes of *Vaccinium* spp. berries. Total phenolics were quantified via the Folin-Ciocalteu method and significant differences between genotypes in total phenolic content are denoted by capital letters. Total monomeric anthocyanins were quantified via the pH differential method and significant differences between genotypes in total monomeric anthocyanin content are denoted by lower case letters. Data are shown as means ± SDs. Significant differences between genotypes were determined using Tukey’s HSD test (*p* < 0.05).

**Figure 4 antioxidants-12-01136-f004:**
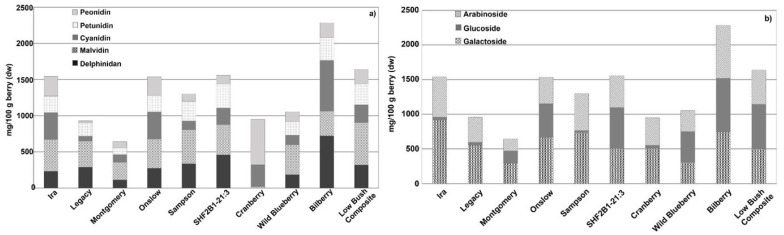
Anthocyanin content of 10 genotypes of *Vaccinium* spp. berries. (**a**) Sum of anthocyanins sharing the same aglycone and (**b**) sum of anthocyanins sharing the same glycosylation pattern. Data are shown as means.

**Figure 5 antioxidants-12-01136-f005:**
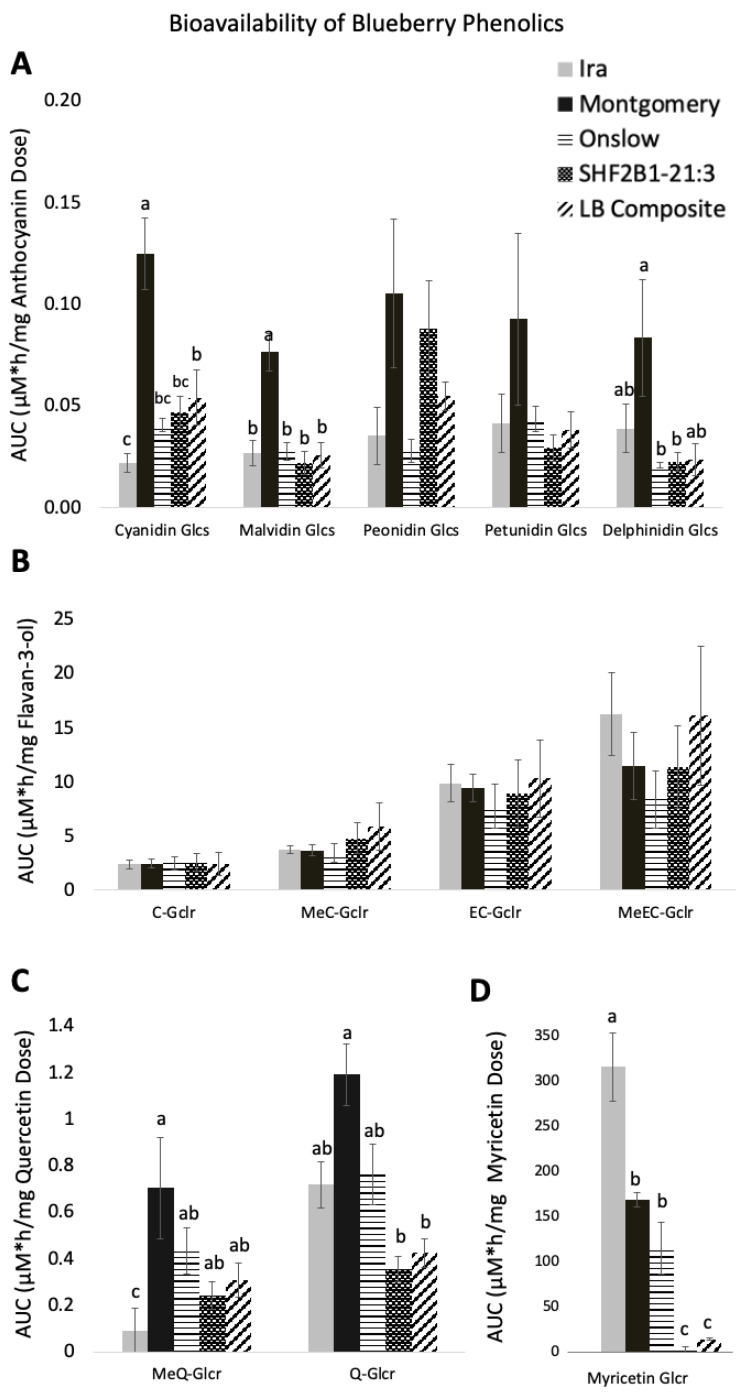
Bioavailability of blueberry polyphenolics. (**A**) anthocyanin glycosides, (**B**) Flavan-3-ol metabolites, (**C**) quercetin metabolites, and (**D**) myricitin metabolites. AUC 0-8h was normalized by blueberry dose. Letters over the bars represent significant differences in bioavailability of poly-phenolics among groups. Letters represent significant differences in bioavailability of polyphenolics among groups.

**Figure 6 antioxidants-12-01136-f006:**
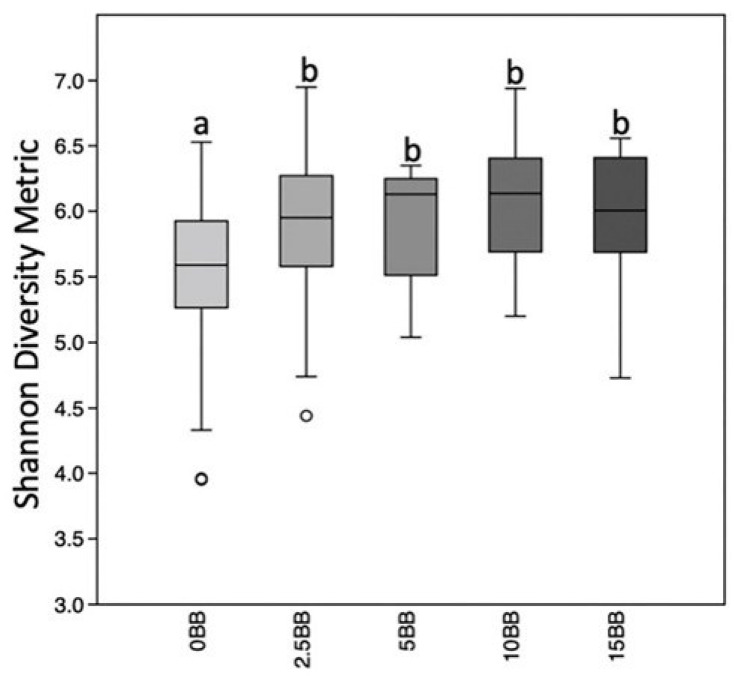
Greater diversity within bacterial communities after blueberry diets. Boxplots representing the medians and upper and lower quartile values of within sample (alpha) diversity using the Shannon diversity metric. Krustal Wallis indicates overall differences is significant (*p* = 0.001) among blueberry dosages and pairwise-tests with 999 permutations significant differences q < 0.05 illustrated by different letters above bars. Outliers are represented by open circles.

**Figure 7 antioxidants-12-01136-f007:**
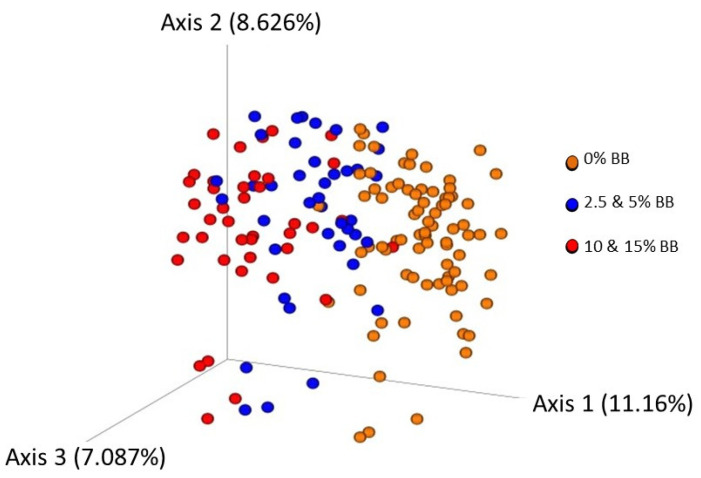
Principal coordinate analysis illustrating differences unweighted Bray Curtis beta diversity metrics among bacterial communities after blueberry diets. Data points colored by blueberry dosages, 2.5 and 5% combined, 10 and 15% combined and no blueberry control. perMANOVA comparisons of samples blueberry dosage, overall differences significant *p* = 0.001 and all pairwise comparisons q = 0.001. Overall and pairwise permDISP comparisons not significant, indicating differences are not due to differences in variation.

**Figure 8 antioxidants-12-01136-f008:**
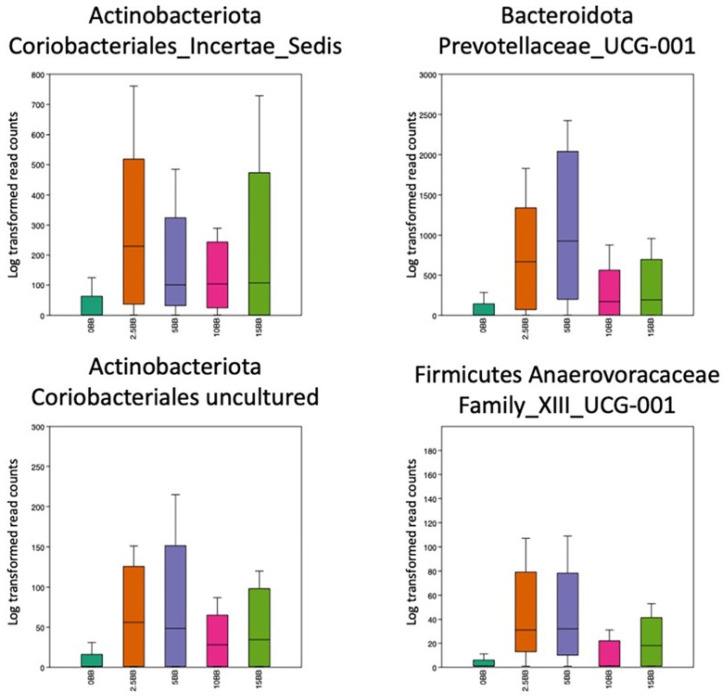
Differentially abundant taxa after different blueberry dosage diets identified using Analysis of Composition of Microbiomes (ANCOM) of microbial communities. Boxplot of medians and SD (standard deviations) illustrate taxa differences with ANCON W values ≥100 and clr > 40.

**Table 1 antioxidants-12-01136-t001:** Polyphenolic compounds and metabolites analyzed and quantified on LC-MS.

Compound	SIR for Quantitation (*m*/*z*)	MRM Transition for Quantitation (*m*/*z*)	Standard Used for Quantification
*Anthocyanins*			
Cyanidin *	287	463 ⟶ 287	Cyanidin-3-*O*-glucoside
Peonidin *	301	463 ⟶ 301	
Delphinidin *	303	465 ⟶ 303	
Petunidin *	317	479 ⟶ 317	
Malvidin *	331	493 ⟶ 331	
*Phenolic Acids ***			
Gallic acid	169		Gallic acid
Chlorogenic acid	353		Chlorogenic acid
*Flavan-3-ols*			
Catechin/Epicatechin	289		Epicatechin
Catechin/Epicatechin-*O*-glucuronide		465 ⟶ 289	
Me-Catechin/Epicatechin-*O*-glucuronide		479⟶ 303	
Procyanidins B1 and B2	577		
*Flavonols*			
Quercetin-3-O-arabinoside	433		Quercetin-3-*O*-glucoside
Quercetin-3-O-galactoside/glucoside	463		
Quercetin-3-*O*-glucuronide	477	477 ⟶ 301	
Quercetin-3-*O*-rhamnoside	447		
Me-Quercetin-3-*O*-glucuronide		491 ⟶ 315	
Quercetin-3-*O*-rutinoside	609		Quercetin-3-*O*-rutinoside
Kaempferol-3-*O*-galactoside/glucoside	447		Kaempferol-3-*O*-rutinoside
Kaempferol-3-*O*-rutinoside	593		
Myricetin-3-*O*-galactoside/glucoside	479		Quercetin-3-*O*-glucoside
Myricetin-3-*O*-glucuronide		493 ⟶ 317	

* Measured 3-*O*-arabinoside, galactoside, and glucoside derivatives, as well as acylated forms. ** Additional phenolics were included in method, but were below the LOD. These included salicylic acid, protocatechuic acid, p-coumaric acid, caffeic acid, ferulic acid, sinapic acid, ellagic acid, resveratrol, piceatannol, pterostilbene, EGC, ECG, and isorhamnetin-3-*O*-glycosides.

**Table 2 antioxidants-12-01136-t002:** Anthocyanin content of 10 genotypes of *Vaccinium* spp. (mg phenolic/100 g berry (dw)), from HPLC-MS analysis ^†^.

		Delphinidin	Malvidin	Cyanidin	Petunidin	Peonidin	Total
Ira	3-*O*-gal	150 ± 3.78 ^d,e^	237 ± 13.2 ^a^	230 ± 6.0 ^b^	142 ± 5.5 ^b^	159 ± 9.3 ^b^	916 ± 38 ^A^
	3-*O*-glu	3.1 ± 0.2 ^f^	14.5 ± 0.5 ^e^	8.5 ± 0.1 ^f^	3.7 ± 0.5 ^e^	15.0 ± 0.6 ^g^	44.8 ± 2.0 ^E^
	3-*O*-ara	78.5 ± 4.3 ^d,e^	190 ± 15.4 ^a^	134 ± 5.9 ^b^	79.7 ± 5.5 ^c,d^	99.7 ± 3.9 ^b^	582 ± 35 ^B^
	Other forms	42.5 ± 5.9 ^e^	42.4 ± 10.1 ^d,e^	40.0 ± 8.6 ^b,c,d^	16.7 ± 4.2 ^e,f^	30.4 ± 6.4 ^c,d^	172 ± 35 ^E^
	Total	274 ± 14 ^ζ^	483 ± 39 ^γ,δ^	412 ± 21 ^β^	242 ± 16 ^ε^	304 ± 20 ^β^	1715
Legacy	3-*O*-gal	184 ± 15.9 ^b,c^	195 ± 12.8 ^b^	41.5 ± 4.0 ^i^	112 ± 9.9 ^c^	17.7 ± 1.4 ^f^	551 ± 44 ^C^
	3-*O*-glu	3.5 ± 0.3 ^f^	7.3 ± 1.4 ^e^	0.2 ± 0.2 ^g^	4.5 ± 1.3 ^e^	0.2 ± 0.2 ^h^	15.6 ± 3.3 ^E^
	3-*O*-ara	101 ± 7.7 ^c^	156 ± 7.9 ^b^	28.6 ± 2.6 ^f^	62.0 ± 3.9 ^e^	14.0 ± 1.2 ^f^	362 ± 23 ^E,F^
	Other forms	62.4 ± 3.9 ^d^	13.0 ± 5.1 ^f^	2.5 ± 0.9 ^f^	5.3 ± 1.6 ^g^	0.5 ± 0.5 ^e^	83.8 ± 12 ^F^
	Total	351 ± 26 ^δ,ε^	372 ± 17 ^ε^	72.8 ± 5.9 ^η^	184 ± 14 ^ζ^	32.4 ± 2.6 ^ζ^	1013
Montgomery	3-*O*-gal	58.8 ± 12.8 ^g^	94.9 ± 15.9 ^d,e^	62.9 ± 14.9 ^g,h^	40.5 ± 8.3 ^f^	33.6 ± 5.4 ^e,f^	291 ± 57 ^D^
	3-*O*-glu	24.3 ± 5.7 ^e^	74.1 ± 11.3 ^d^	20.1 ± 4.2 ^e^	28.0 ± 5.6 ^d^	33.9 ± 5.6 ^f^	180 ± 32 ^D^
	3-*O*-ara	28.5 ± 6.0 ^g^	73.1 ± 8.3 ^e^	26.0 ± 4.7 ^f^	22.3 ± 3.9 ^g^	21.5 ± 2.4 ^e,f^	171 ± 25 ^G^
	Other forms	16.2 ± 1.3 ^g^	11.1 ± 2.2 ^f^	0.3 ± 0.6 ^f^	4.3 ± 0.5 ^g^	5.1 ± 1.2 ^e^	37.1 ± 5.8 ^F^
	Total	128 ± 26 ^η^	253 ± 34 ^ζ^	109 ± 23 ^ζ,η^	95 ± 18 ^η^	94 ± 13 ^ε^	679
Onslow	3-*O*-gal	139 ± 2.6 ^e^	142 ± 1.3 ^c^	191 ± 2.7 ^c^	97.4 ± 1.7 ^c,d,e^	98.3 ± 3.0 ^c^	668 ± 11 ^B^
	3-*O*-glu	69.7 ± 1.0 ^d^	156 ± 1.8 ^c^	85.2 ± 1.1 ^b^	78.5 ± 0.3 ^c^	97.4 ± 0.4 ^b^	486 ± 4.6 ^C^
	3-*O*-ara	64.1 ± 0.9 ^e,f^	107 ± 0.9 ^d^	100 ± 1.3 ^c^	50.7 ± 0.3 ^f^	60.7 ± 1.6 ^c^	382 ± 5.1 ^E^
	Other forms	30.8 ± 2.2 ^f^	37.3 ± 5.4 ^e^	23.9 ± 5.3 ^e^	21.1 ± 2.9 ^e^	33.9 ± 5.2 ^c^	147 ± 21 ^E^
	Total	304 ± 3.2 ^ε,ζ^	442 ± 3.4 ^δ^	400 ± 1.8 ^β^	248 ± 2.6 ^ε^	290 ± 5.6 ^β^	1684
Sampson	3-*O*-gal	203 ± 6.9 ^b^	245 ± 2.9 ^a^	67.1 ± 6.5 ^f,g^	165 ± 3.1 ^a^	57.0 ± 19.7 ^d,e^	737 ± 39 ^B^
	3-*O*-glu	5.0 ± 0.4 ^f^	15.3 ± 1.2 ^e^	1.22 ± 0.2 ^f,g^	5.3 ± 0.3 ^e^	2.1 ± 1.2 ^h^	28.9 ± 3.4 ^E^
	3-*O*-ara	129 ± 4.2 ^b^	209 ± 1.7 ^a^	53.3 ± 5.7 ^e^	104 ± 0.5 ^a^	36.9 ± 14.5 ^d^	532 ± 27 ^B,C^
	Other forms	62.1 ± 3.0 ^d^	61.4 ± 2.0 ^d^	2.0 ± 0.6 ^f^	16.3 ± 0.8 ^e,f^	4.4 ± 0.6 ^e^	146 ± 7.0 ^E^
	Total	399 ± 14 ^δ^	531 ± 3.7 ^γ^	124 ± 13 ^ε,ζ^	291 ± 2.8 ^δ^	100 ± 36 ^ε^	1444
SHF2 B1-21:3	3-*O*-gal	166 ± 9.1 ^c,d^	108 ± 3.8 ^d^	93.0 ± 3.9 ^e^	102 ± 6.1 ^c,d^	39.0 ± 1.5 ^d,e,f^	508 ± 24 ^C^
	3-*O*-glu	159 ± 11.0 ^b^	173 ± 10.4 ^b^	65.8 ± 2.7 ^c^	143 ± 8.2 ^a^	48.5 ± 2.3 ^d,e^	589 ± 35 ^B^
	3-*O*-ara	134 ± 6.4 ^b^	136 ± 4.3 ^c^	72.5 ± 2.2 ^d^	86.7 ± 3.0 ^b,c^	30.6 ± 1.2 ^d,e^	461 ± 17 ^D^
	Other forms	130 ± 5.7 ^b^	223 ± 4.3 ^b^	43.5 ± 3.1 ^b^	111 ± 2.2 ^a^	43.3 ± 2.2 ^b,c^	551 ± 17 ^B^
	Total	589 ± 26 ^β^	641 ± 22 ^β^	275 ± 8.2 ^δ^	442 ± 18 ^α^	162 ± 6.0 ^δ^	2108
Cranberry	3-*O*-gal	nd	9.0 ± 0.4 ^f^	157 ± 6.9 ^d^	trace	342 ± 13.3 ^a^	507 ± 21 ^C^
	3-*O*-glu	nd	trace	4.6 ± 0.2 ^f,g^	nd	41.7 ± 3.5 ^e^	46.3 ± 3.7 ^E^
	3-*O*-ara	nd	11.0 ± 0.3 ^f^	139 ± 1.2 ^b^	trace	247 ± 4.7 ^a^	396 ± 6.3 ^E^
	Other forms	90.4 ± 2.8 ^c^	0.5 ± 0.5 ^f^	41.7 ± 8.7 ^b,c^	32.6 ± 0.2 ^d^	94.8 ± 21.1 ^a^	260 ± 33 ^D^
	Total	90.4 ± 2.8 ^η^	20.5 ± 0.4 ^η^	342 ± 2.6 ^γ^	32.6 ± 0.2 ^θ^	725 ± 0.2 ^α^	1210
Wild Blueberry	3-*O*-gal	58.8 ± 3.5 ^g^	113 ± 5.4 ^d^	42.7 ± 1.6 ^h,i^	52.4 ± 2.8 ^f^	40.7 ± 1.1 ^d,e,f^	307 ± 14 ^D^
	3-*O*-glu	78.7 ± 3.7 ^d^	170 ± 6.5 ^b,c^	50.4 ± 3.2 ^d^	86.6 ± 4.8 ^c^	55.9 ± 2.4 ^d^	442 ± 21 ^C^
	3-*O*-ara	46.5 ± 2.5 ^f,g^	132 ± 6.4 ^c^	41.3 ± 2.5 ^e^	46.1 ± 2.1 ^f^	39.7 ± 1.5 ^d^	305 ± 15 ^F^
	Other forms	93.2 ± 5.3 ^c^	187 ± 14.0 ^c^	27.2 ± 2.1 ^d,e^	47.6 ± 3.8 ^c^	58.6 ± 4.3 ^b^	414 ± 29 ^C^
	Total	277 ± 14 ^ζ^	602 ± 31 ^β^	162 ± 9.2 ^ε^	233 ± 13 ^ε^	195 ± 8.8 ^γ,δ^	1468
Bilberry	3-*O*-gal	258 ± 9.4 ^a^	79.0 ± 3.0 ^e^	269 ± 11.5 ^a^	92.1 ± 4.3 ^d,e^	45.1 ± 1.2 ^d,e^	744 ± 29 ^B^
	3-*O*-glu	186 ± 9.1 ^a^	162 ± 3.0 ^b,c^	182 ± 6.3 ^a^	131 ± 6.1 ^a,b^	114 ± 2.9 ^a^	775 ± 27 ^A^
	3-*O*-ara	277 ± 13.4 ^a^	98.9 ± 1.5 ^d^	253 ± 8.4 ^a^	93.3 ± 3.6 ^b^	43.1 ± 1.0 ^d^	765 ± 28 ^A^
	Other forms	20.1 ± 1.1 ^f,g^	13.9 ± 3.2 ^f^	28.8 ± 5.9 ^c,d,e^	13.7 ± 2.6 ^f^	9.0 ± 2.1 ^d,e^	85.6 ± 15 ^F^
	Total	741 ± 33 ^α^	354 ± 7.8 ^ε^	733 ± 28 ^α^	330 ± 15 ^γ^	211 ± 5.5 ^γ^	2369
Lowbush composite	3-*O*-gal	107 ± 1.0 ^f^	163 ± 1.7 ^c^	85.4 ± 1.2 ^e,f^	84.1 ± 0.6 ^e^	62.9 ± 0.8 ^d^	502 ± 5.2 ^C^
	3-*O*-glu	125 ± 1.2 ^c^	228 ± 2.4 ^a^	85.2 ± 0.9 ^b^	126 ± 1.0 ^b^	77.7 ± 0.6 ^c^	641 ± 6.1 ^B^
	3-*O*-ara	86.6 ± 2.1 ^c,d^	194 ± 4.6 ^a^	79.0 ± 1.0 ^d^	76.5 ± 2.9 ^d^	60.7 ± 0.8 ^c^	497 ± 11 ^C,D^
	Other forms	168 ± 5.0 ^a^	334 ± 7.3 ^a^	65.4 ± 2.1 ^a^	96.5 ± 1.9 ^b^	96.0 ± 3.1 ^a^	760 ± 19 ^A^
	Total	487 ± 6.3 ^γ^	918 ± 9.2 ^α^	315 ± 3.9 ^γ,δ^	383 ± 3.5 ^β^	297 ± 3.7 ^β^	2400

^†^ Lowercase letters represent significant differences in individual anthocyanins between different *Vaccinium* spp. genotypes; capital letters represent significant differences in anthocyanins with the same glycosylation/acylation pattern between genotypes; Greek letters represent significant differences in anthocyanins sharing the same aglycone between genotypes. Significant differences between genotypes were determined using Tukey’s HSD test (*p* < 0.05). “Other forms” represents the sum of anthocyanins with acylated side chains. “Total” represents the sum of all anthocyanins with the same parent anthocyanidin. “Trace” indicates compounds that were detected but below the LOQ, while “nd” indicates phenolics that were not detected.

**Table 3 antioxidants-12-01136-t003:** Phenolic acids and flavan-3-ols in 10 genotypes of *Vaccinium* spp. (mg phenolic/100 g berry (dw)) ^†^.

	Phenolic Acids		Flavan-3-ols			
	Gallic Acid	Chlorogenic Acid	Catechin	Epicatechin	Procyanidin B1	Procyanidin B2
Ira	0.70 ± 0.09 ^b,c,d^	755 ± 46.7 ^a^	15.9 ± 1.28 ^a^	1.07 ± 0.25 ^d,e^	8.27 ± 0.96 ^a^	0.75 ± 0.31 ^e^
Legacy	0.52 ± 0.04 ^c,d^	520 ± 58.5 ^b^	8.11 ± 0.67 ^c^	0.71 ± 0.27 ^e^	4.02 ± 0.23 ^c^	2.51 ± 0.44 ^d,e^
Montgomery	0.47 ± 0.03 ^c,d^	493 ± 48.9 ^b^	12.8 ± 0.29 ^b^	1.63 ± 0.07 ^c,d,e^	6.56 ± 0.49 ^b^	2.03 ± 0.23 ^e^
Onslow	0.57 ± 0.19 ^c,d^	496 ± 30.5 ^b^	15.2 ± 0.57 ^a^	3.24 ± 0.09 ^c^	7.17 ± 0.29 ^a,b^	4.23 ± 0.24 ^c,d^
Sampson	0.92 ± 0.09 ^b,c^	269 ± 9.03 ^c^	3.90 ± 0.16 ^d^	0.41 ± 0.28 ^e^	0.91 ± 0.19 ^d^	0.89 ± 0.44 ^e^
SHF2B1-21:3	0.66 ± 0.04 ^b,c,d^	492 ± 26.4 ^b^	6.97 ± 0.10 ^c^	1.89 ± 0.88 ^c,d,e^	3.23 ± 0.26 ^c^	2.67 ± 0.62 ^d,e^
Cranberry	0.38 ± 0.08 ^d^	20.3 ± 0.64 ^d^	1.93 ± 0.07 ^e^	19.3 ± 0.82 ^a^	trace	13.8 ± 1.49 ^b^
Wild BB	0.52 ± 0.02 ^c,d^	566 ± 31.6 ^b^	7.64 ± 0.78 ^c^	3.04 ± 0.32 ^c,d^	6.21 ± 0.31 ^b^	4.62 ± 0.48 ^c,d^
Bilberry	2.04 ± 0.37 ^a^	104 ± 3.58 ^c,d^	0.06 ± 0.06 ^f^	10.9 ± 1.75 ^b^	trace	19.3 ± 1.33 ^a^
LB composite	1.11 ± 0.32 ^b^	740 ± 25.1 ^a^	8.02 ± 0.43 ^c^	1.67 ± 0.64 ^c,d,e^	7.02 ± 0.53 ^a,b^	5.44 ± 0.62 ^c^

^†^ Letters represent significant differences in phenolics between genotypes. Significant differences between genotypes were determined using Tukey’s HSD test (*p* < 0.05). “Trace” indicates compounds that were detected but below the LOQ.

**Table 4 antioxidants-12-01136-t004:** Flavonols in 10 genotypes of *Vaccinium spp.* (mg phenolic/100 g berry (dw)) ^†^.

	Quer 3-*O*-arb	Quer 3-*O*-rham	Quer 3-*O*-glcs	Quer 3-*O*-gcnd	Quer 3-*O*-rut	Total Quer	Kaemp 3-*O*-glcs	Kaemp 3-*O*-rut	Myr 3-*O*-glcs
Ira	0.34 ± 0.14 ^f^	58.1 ± 9.21 ^b^	26.1 ± 3.18 ^d^	8.92 ± 0.34 ^c,d^	14.3 ± 1.30 ^c,d,e^	108 ± 11.7 ^e^	0.09 ± 0.08 ^d^	trace	2.20 ± 0.12 ^e^
Legacy	28.1 ± 7.30 ^d^	24.7 ± 6.50 ^c^	249 ± 29.1 ^a^	1.14 ± 0.48 ^d^	9.18 ± 2.16 ^d,e,f^	313 ± 43.7 ^b,c^	3.35 ± 1.03 ^a^	0.57 ± 0.31 ^b,c^	19.0 ± 1.40 ^c,d^
Montgomery	17.3 ± 1.40 ^d,e,f^	27.7 ± 1.35 ^c^	23.9 ± 2.83 ^d^	6.04 ± 0.59 ^d^	17.6 ± 1.15 ^c,d^	92.5 ± 5.48 ^e^	0.59 ± 0.06 ^c,d^	0.71 ± 0.11 ^b,c^	4.86 ± 0.10 ^d,e^
Onslow	26.3 ± 4.12 ^d^	47.8 ± 4.76 ^b^	33.3 ± 4.14 ^d^	15.5 ± 0.44 ^c^	7.68 ± 1.85 ^e,f^	131 ± 5.70 ^e^	0.73 ± 0.12 ^c,d^	0.34 ± 0.19 ^c^	2.35 ± 0.54 ^e^
Sampson	92.7 ± 8.20 ^b^	113 ± 6.15 ^a^	124 ± 13.0 ^c^	trace	31.3 ± 3.35 ^a^	360 ± 22.4 ^a,b^	1.10 ± 0.12 ^c,d^	1.77 ± 0.36 ^a^	26.1 ± 4.18 ^c^
SHF2B1-21:3	20.6 ± 2.35 ^d,e^	13.9 ± 2.14 ^c,d^	156 ± 16.6 ^c^	trace	4.81 ± 0.74 ^f^	194 ± 19.1 ^d^	0.64 ± 0.07 ^c,d^	trace	52.1 ± 15.7 ^a,b^
Cranberry	136 ± 11.0 ^a^	21.1 ± 3.05 ^c^	206 ± 15.8 ^b^	trace	trace	363 ± 22.6 ^a,b^	0.72 ± 0.26 ^c,d^	nd	58.7 ± 9.74 ^a^
Wild BB	26.9 ± 2.12 ^d^	20.7 ± 1.94 ^c^	161 ± 21.9 ^c^	40.3 ± 6.84 ^b^	20.9 ± 3.06 ^b,c^	269 ± 25.5 ^c^	1.59 ± 0.30 ^b,c^	1.32 ± 0.22 ^a,b^	22.4 ± 3.80 ^c,d^
Bilberry	7.37 ± 1.26 ^e,f^	1.09 ± 0.21 ^d^	47.5 ± 6.38 ^d^	38.5 ± 1.40 ^b^	trace	94.5 ± 8.75 ^e^	trace	trace	35.3 ± 8.01 ^b,c^
LB composite	64.1 ± 12.4 ^c^	54.6 ± 7.43 ^b^	210 ± 9.01 ^a,b^	60.5 ± 4.19 ^a^	29.0 ± 7.50 ^a,b^	418 ± 20.6 ^a^	2.28 ± 0.30 ^a,b^	2.00 ± 0.51 ^a^	27.8 ± 1.52 ^c^

^†^ Letters represent significant differences in phenolics between genotypes. Significant differences between genotypes were determined using Tukey’s HSD test (*p* < 0.05). “Trace” indicates compounds that were detected but below the LOQ, “nd” indicates phenolics that were not detected, and “total quer” represents the total of all glycosides containing quercetin as the parent flavonol. Abbreviations: Quer = quercetin; arb = arabinoside; rham = rhamnoside; glcs = glycosides (galactoside + glucoside); gcnd = glucuronide; rut = rutinoside; Kaemp = kaempferol; Myr = myricetin.

## Data Availability

Data will be made available upon request to senior author. Microbiome sequence data are deposited in NCBI-SRA https://www.ncbi.nlm.nih.gov/ under accession number BioProject ID: PRJNA954427. The data on composition of blueberries were collected at the North Carolina State University Piedmont Research Station (PRS), Salisbury, NC, USA (https://cals.ncsu.edu/food-bioprocessing-and-nutrition-sciences/). More information about the data and its use is available from coauthors Mary Ann Lila and Mario G. Ferruzzi.

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
