# Peer review of "Crop, Host, and Gut Microbiome Variation Influence Precision Nutrition: An Example of Blueberries"

_antioxidants, 2023, doi:10.3390/antiox12051136_

Round 1

Reviewer 1 Report

The authors make a very interesting, extensive and detailed study. The authors have generated a lot of data and have done a detailed statistical analysis. However, they should be more specific in terms of the results of their analysis: better genotypes, more bioavailability and microbiota of the different genotypes and doses, etc.

Both the abstract and the conclusions are very general. They should include the most relevant results in terms of genotypes, doses, bioavailability and microbiota.

Given the interest that the data could have for future improvement work, it would be interesting if they were deposit data in a trusted repository and provide open access, such as zenodo (https://zenodo.org/)

2. Materials and Methods

Line 84  2.1. . Diversity of anthocyanin profiles across blueberry genotypes

Line 152 Why were 20 rats used? From my point of view, the study would have been improved by using a larger number of rats or reducing the number of treatments.

Line 158. It would be advisable to include a diagram of the different treatments carried out on the animals.

3. Results

Line 337: Figure 3 is of low quality.

Line 347. Bioavailability of blueberry metabolites. This is one of the most interesting points of the work. Something more concrete should be commented on the results obtained in the conclusions and abstract.

From page 13 the lines are not indicated.

3.4. Dietary blueberry effects on the gut microbiome. 3.4. Dietary blueberry effects on the gut microbiome. More of the results obtained should be specified in the abstract and in the conclusions.

4. Discussion

A large amount data has been generated in this study. However, the authors do not work with these results to provide new information. They are not very specific both in the discussion and in the conclusions. What is the contribution of this study? What genotype and dose of blueberries are most effective? What genotypes and in what doses are more bioavailable? What genotypes and in what doses improve the microbioma?.

Conclusions

Very general conclusions. The authors must rewrite the conclusions including the specific contributions that are extracted from the large amount of data that has been generated in this work.

Author Response

RESPONSE TO REVIEWERS

We thank the reviewers for their constructive feedback.

Reviewer 1:

The authors make a very interesting, extensive and detailed study. The authors have generated a lot of data and have done a detailed statistical analysis. However, they should be more specific in terms of the results of their analysis: better genotypes, more bioavailability and microbiota of the different genotypes and doses, etc.

Both the abstract and the conclusions are very general. They should include the most relevant results in terms of genotypes, doses, bioavailability and microbiota.

Response: We agree. The abstract was expanded to include: Principal component analysis was used to select blueberry genotypes that varied in anthocyanin profiles. Total phenolic content did not predict bioavailability of polyphenolic compounds in rats. A range in bioavailability was observed in individual polyphenolic compounds across genotypes.

The conclusions were expanded to include: PCA of 17 anthocyanins from 267 genotypes allowed selection of a few genotypes with widely differing profiles for further characterization. The high bioavailability of anthocyanins and phenolics in certain genotypes is a unique contribution of this work. Further, evidence of a gut microbiome response to blueberry dose was observed.

Given the interest that the data could have for future improvement work, it would be interesting if they were deposit data in a trusted repository and provide open access, such as zenodo (https://zenodo.org/)

Response: This was added to the data sharing section: The data on composition of blueberries were collected at the North Carolina State University Piedmont Research Station (PRS), Salisbury, North Carolina, USA (https://cals.ncsu.edu/food-bioprocessing-and-nutrition-sciences/).  More information about the data and its use is available from coauthors Mary Ann Lila and Mario G. Ferruzzi.  

  1. Materials and Methods

Line 84  2.1. . Diversity of anthocyanin profiles across blueberry genotypes

Response:  Was there a comment/suggestion?

Line 152 Why were 20 rats used? From my point of view, the study would have been improved by using a larger number of rats or reducing the number of treatments.

Response:  There is always a tradeoff between sample size and number of treatments. Our sample size of 4 rats per treatment gave us the power to distinguish important differences.

Line 158. It would be advisable to include a diagram of the different treatments carried out on the animals.

Response: A new Figure 1 of study design was added.

  1. Results

Line 337: Figure 3 is of low quality.

Response: This figure was replaced.

Line 347. Bioavailability of blueberry metabolites. This is one of the most interesting points of the work. Something more concrete should be commented on the results obtained in the conclusions and abstract.

Response: We agree. The abstract was expanded to include: Principal component analysis was used to select blueberry genotypes that varied in anthocyanin profiles. Total phenolic content did not predict bioavailability of polyphenolic compounds in rats. A range in bioavailability was observed in individual polyphenolic compounds across genotypes.

From page 13 the lines are not indicated.

Response: Lines are now numbered.

3.4. Dietary blueberry effects on the gut microbiome. 3.4. Dietary blueberry effects on the gut microbiome. More of the results obtained should be specified in the abstract and in the conclusions.  

Response: The abstract and conclusions were expanded as discussed above.

  1. Discussion

A large amount data has been generated in this study. However, the authors do not work with these results to provide new information. They are not very specific both in the discussion and in the conclusions. What is the contribution of this study? What genotype and dose of blueberries are most effective? What genotypes and in what doses are more bioavailable? What genotypes and in what doses improve the microbioma?.

Response: The conclusions were expanded as described above. The results are quite extensive in reporting different outcomes with genotype and dose as available. We did not compare genotype effects on the gut microbiome in this rat study, but we did compare 3 Rabbiteye genotypes in a previously reported mouse study.  We added this sentence to the limitations: Genotype comparisons should be made for the gut microbiome as was done in our previous report of a mouse study [49].

Conclusions

Very general conclusions. The authors must rewrite the conclusions including the specific contributions that are extracted from the large amount of data that has been generated in this work.

Response: The conclusions were expanded to include: PCA of 17 anthocyanins from 267 genotypes allowed selection of a few genotypes with widely differing profiles for further characterization. The high bioavailability of anthocyanins and phenolics in certain genotypes is a unique contribution of this work. Further, evidence of a gut microbiome response to blueberry dose was observed.

Reviewer 2 Report

The article studies the relationship between the phenolic composition of blueberry, the bioavailability of phenols and their effect on the intestinal microbiome using a rat model.

The experimental strategy is correct and the methodology used adequate. The results are clearly presented and the conclusions are consistent with them.

Just a few minor issues:

Line 87: Probably the abbreviation should be “NCSU-PRS” instead of “PRS”

Line 11-112: Please explain how did you prepare and purify the blueberry extracts.

Line 164: Probably “suspended” would be more appropriated than “dissolved”

Line 241: Please change “2.3.2” to “2.4.2”.

Line 266: Please change “2.3.3” to “2.4.3”

Page 20: Please chage “3.3.6” to “3.4.6”

Author Response

Reviewer 2:

The article studies the relationship between the phenolic composition of blueberry, the

bioavailability of phenols and their effect on the intestinal microbiome using a rat model.

The experimental strategy is correct and the methodology used adequate. The results

are clearly presented and the conclusions are consistent with them.

Just a few minor issues:

Line 87: Probably the abbreviation should be “NCSU-PRS” instead of “PRS”

Response:  Done.

Line 11-112: Please explain how did you prepare and purify the blueberry extracts.

Response: This was added to the beginning of section 2.22: Lyophilized whole blueberries were ground into a fine powder using a spice grinder. Phenolics were then extracted, purified, and analyzed in triplicate for total phenolics via the Folin method, as described elsewhere [23,24].

Line 164: Probably “suspended” would be more appropriated than “dissolved”

Response:  Done.

Line 241: Please change “2.3.2” to “2.4.2”.

Response:  Done.

Line 266: Please change “2.3.3” to “2.4.3”

Response:  Done.

Page 20: Please change “3.3.6” to “3.4.6”

Response:  The section was removed.

Reviewer 3 Report

I guess eventually we would have precision nutrition, although I was not expecting it to present in this way.  Actually, I am reviewing a second paper that uses this terminology to justify the study.   Much of study is fairly well performed.  There is one major short fall in the results and discussion.

Minor- Section 2.22, line 111; what was the initial extraction process?

Figure 1- can we have a more annotated version of the figure, indicating the locations the selected groups?

Figure 5-Greater diversity within(g)

Major-Figure 8 and related results presentation and discussion;  Fig 8 fails to convey anything to me. Can something else be substituted in place of this? Maybe, 3 correlation graphs?  I certainly can not make much out of the impact of blueberries on the Ca retention from this.  And, this is the major short fall of the paper-the discussion omits real comment on the Ca retention results

Author Response

Reviewer 3:

I guess eventually we would have precision nutrition, although I was not expecting it to present in this way.  Actually, I am reviewing a second paper that uses this terminology to justify the study.   Much of study is fairly well performed.  There is one major short fall in the results and discussion.

Minor- Section 2.22, line 111; what was the initial extraction process?

Response: This was added to the beginning of section 2.22: Lyophilized whole blueberries were ground into a fine powder using a spice grinder. Phenolics were then extracted, purified, and analyzed in triplicate for total phenolics via the Folin method, as described elsewhere [23,24].

Figure 1- can we have a more annotated version of the figure, indicating the locations the selected groups?

Response:  As suggested, the figure was replaced with one that indicates locations of the selected genotype.

Figure 5-Greater diversity within(g)

Response:  Done

Major-Figure 8 and related results presentation and discussion;  Fig 8 fails to convey anything to me. Can something else be substituted in place of this? Maybe, 3 correlation graphs?  I certainly can not make much out of the impact of blueberries on the Ca retention from this.  And, this is the major short fall of the paper-the discussion omits real comment on the Ca retention results 

Response: Fig 8 was removed.

Round 2

Reviewer 3 Report

The revised paper look good to me.